# Triple Alignment Strategies for
# Zero-shot Phrase Grounding under Weak Supervision

## ABSTRACT

*Phrase Grounding*, i.e., PG aims to locate objects referred by noun phrases. Recently, PG under weak supervision (i.e., grounding without region-level annotations) and zero-shot PG (i.e., grounding from seen categories to unseen ones) are proposed, respectively. However, for real-world applications these two approaches are limited due to slight annotations and numerable categories during training. In this paper, we propose a framework of zero-shot PG under weak supervision. Specifically, our PG framework is built on triple alignment strategies. Firstly, we propose a region-text alignment (RTA) strategy to build region-level attribute associations via CLIP. Secondly, we propose a domain alignment (DomA) strategy by minimizing the difference between distributions of seen classes in the training and those of the pre-training. Thirdly, we propose a category alignment (CatA) strategy by considering both category semantics and region-category relations. Extensive experiment results show that our proposed PG framework outperforms previous zero-shot methods and achieves competitive performance compared with existing weakly-supervised methods. The code and data will be publicly available at GitHub after double-blind phase.

## KEYWORDS

Vision and language, Phrase grounding, Weakly supervised, Zero-shot, Vision-language pre-training.

## 1 INTRODUCTION

*Phrase Grounding* (i.e., PG) [48] aims to locate objects referred by noun phrases. The PG task could be beneficial for various downstream works, such as image captioning [8, 31, 56], vision navigation [3, 16, 24, 51, 52], visual question answering [7, 10, 50, 54], and other multi-modal researches [18, 19, 23, 28–30, 33, 44, 49].

Recently, some works on PG under weak supervision [11] have been proposed. The motivation is to alleviate the large annotation cost of bounding boxes(bbox). PG models under weak supervision are required to learn only from image-phrase pairs but without bounding boxes. To address this challenge, various approaches using visual detectors [9, 14, 32, 34] and constructing auxiliary tasks [1, 2, 12, 17, 41, 42, 55] are proposed. These works achieve significant performance. However, due to limited seen categories during training, they are difficult to apply in zero-shot settings.

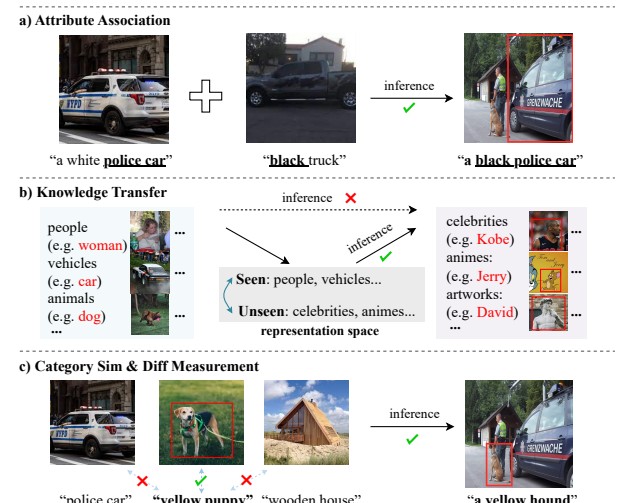

**Figure 1: Three challenges in zero-shot PG under weak supervision are illustrated. a) The attributes association of seen categories with those of unseen categories. b) The knowledge transfer from seen categories to unseen ones via a representation space for prediction. c) The measurement of similarities and differences among categories.**

Very recently, some works propose the task of zero-shot PG. Various approaches using zero-shot learning [5, 40] and Vision-Language Pre-training (VLPs) models [4, 22, 43, 45, 58] are proposed. The former approaches assume that the entities of phrases in the training set have limited categories (i.e., seen categories) and differ from those (i.e., unseen categories) in the testing set. To bridge the gap, these works leverage semantic information shared across all categories, achieving good performance. However, these works still require collecting large-scale bounding box annotations for seen categories. The latter VLP-based methods ground objects on new data without being fine-tuned. However, these methods primarily showcase the generality of VLPs for various vision-language tasks, not focusing on PG. In addition, these VLP-based methods suffer from the collecting of enormous training data.

To summarize, existing works have not addressed the challenges of PG in weak supervision and zero-shot settings, simultaneously. In effect, for real-world applications, the grounding models are required to fit images without bounding box annotations and to effectively generalize from limited seen categories to unseen ones. Motivated by this observation, we propose a framework of zero-shot PG under weak supervision.

Here, three questions arise naturally for zero-shot PG under weak supervision by our observations. **Firstly**, how to associate attributes of seen categories with those of unseen categories? Take

an example in Figure 1 a). Suppose that two classes including "white police cat" and "black truck" have been seen before, and the model is required to ground the unseen class "black police car". To perform inference correctly, the semantic information of "black" and "police car" should be transferred to their corresponding visual regions. To this end, the model is required to design a region-text alignment strategy. **Secondly**, how to transfer knowledge learned from seen categories to unseen categories for the model's prediction? As shown in Figure 1 b), during training, the model can only observe the data of limited classes, such as people, vehicles, and animals. We expect that the model could learn a domain-invariant representation space even being trained on seen categories. Thus, the model can take advantage of such invariance for grounding unseen categories, such as celebrities, animes, and artworks. To this end, we need to design a domain alignment strategy. **Thirdly**, how to measure the similarities and differences among seen categories and those of unseen ones? As shown in Figure 1 c), for one thing, "yellow hound" is related to "yellow puppy", but not to "police car". Therefore, phrases can be used to categorize referred objects. For another, correct visual-textual relations can help in distinguishing categories. For "yellow puppy", we reduce its distance to the corresponding region and increase that to irrelevant regions. Then the model will identify the category and ground the region of a similar phrase such as "yellow hound". To this end, we need to design a category alignment strategy. *To emphasize*, the weakly supervised setting requires that the training datasets do not provide accurate location annotations in images. This makes it more difficult for PG models to generalize in the zero-shot setting.

In this paper, we propose a novel PG framework (Figure 2) using triple alignment strategies. **Firstly**, we design a region-text alignment (RTA) strategy to build region-level attribute associations based on Contrastive Language-Image Pre-Training (CLIP). Specifically, we extract region-level visual semantics and gradient maps using CLIP [37] for given phases. Each region-level visual semantics corresponds to a certain text embedding. Then, we use a patch-level gradient map to refine the CLIP-based heatmap. Here, the CLIP-based heatmap is used as a pseudo-label. **Secondly**, we propose a domain alignment (DomA) strategy to transfer knowledge learned from seen classes. Specifically, we align the grounding-related features to those of the pre-trained model through learning a domain-invariant space. A domain alignment loss is designed to adaptively adjust the grounding-related features. **Thirdly**, we design a category alignment (CatA) strategy to distinguish grounding-region categories. Inspired by the class activation method [27], we discriminate the categories based on the phrase embeddings. To construct accurate visual-textual relationships, we use CLIP to measure the similarity of image regions and phrases. Our strategy considers both category semantics and region-category relations.

In summary, our main contributions are three-fold.

**1)** We propose a novel PG framework with zero-shot under weak supervision. To the best of our knowledge, we are the first to study PG under the two settings, simultaneously.

**2)** We propose triple alignment strategies for PG framework. First, we design RTA strategy to learn region-level attribute associations. Second, we propose DomA strategy to learn domain-invariant representation space. Third, we design CatA strategy to help the network distinguish categories.

**3)** We conduct extensive experiments on benchmark datasets. The results consistently show that our approach significantly outperforms existing zero-shot methods, and achieves competitiveness compared with other weakly-supervised methods.

## 2 RELATED WORK

### 2.1 Weakly-supervised PG

Under weak supervision, PG models can solely learn from image-phrase pairs during the training. To address the challenge, detector-based works use visual detectors and choose the correct proposals that are highly related to the corresponding phrases. For example, Datta et al. [9] align captions with caption-conditioned image representations. Lu et al. [34] combine diverse vision-and-language tasks to improve performance. Gupta et al. [14] associate image regions with caption words by maximizing mutual information. Liu et al. [32] design a relation-aware instance refinement module to construct the phrase-object relations. However, the performances of these methods heavily depend on the quality of visual detectors.

Moreover, the other methods design auxiliary tasks for PG, such as intra-modal classifications and inter-modal alignments. Javed et al. [17] use an attention mechanism to share nouns in captions among images having a common visual region. Zhang et al. [55] propose contrastive attention for task-specific attention maps. Akbari et al. [1] designs a multi-level common semantic space by mapping visual and textual features for phrase grounding. Arbelle et al. [2] use source separation techniques to ground the referred entities in pixel-level. Shaharabany et al. [41] propose WWbl, which learns to create a phrase-related foreground with the CLIP explainability. Recently, Shaharabany and Wolf [42] employ a layer-wise relevance propagation method to integrate relevancy and gradient information. Gomel et al. [12] design a joint learning approach to train the grounding model and an object detector, simultaneously. Nevertheless, all previous weakly-supervised PG methods struggle on unseen categories due to the limited training data.

### 2.2 Zero-shot PG

Zero-shot PG aims to predict the bounding boxes for images of unseen categories but is trained only on those of limited seen categories. Currently, there exist two main approaches, namely zero-shot learning-based and VLP-based methods. In the former methods, Sadhu et al. [40] leverage the supervision of image captions, class names, and bounding boxes, and perform zero-shot predictions via mapping the visual representations and three types of annotations. Chen et al. [5] leverage object priors shared across all categories. The latter VLP-based methods ground phrases on new data without being fine-tuned. Recently, GAE [4] was proposed, advancing VLP-based method for PG. Although this method activates the most discriminative location, the grounding box cannot accurately cover the object. Li et al. [22] use super-pixels technique to generate high-resolution feature maps for phrase grounding. Zhou et al. [58] modify the image encoder of CLIP by transforming the value embedding layer to handle pixel-level predictions. Subramanian et al. [45] use the pre-trained detector to generate a set of proposals and then use CLIP to select the best association between the query and proposals. However, these works either

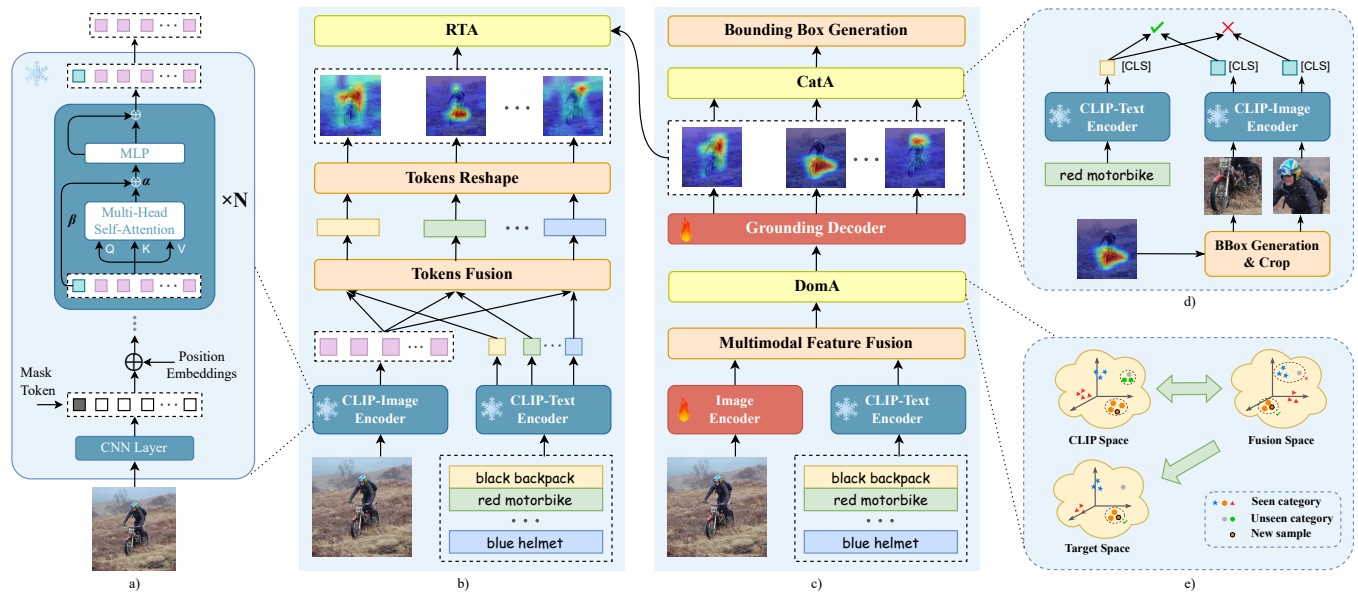

**Figure 2: Our PG framework is depicted. b) CLIP-based module extracts region-level image semantics, and fuses it with text embedding to generate the CLIP-based heatmaps. c) The grounding module consists of bi-modal encoders and one grounding decoder. a) CLIP-based image encoder. e) DomA strategy is used for aligning feature distributions of grounding module and those of CLIP-based module. d) CatA strategy is used for distinguishing grounding-region categories by aligning object and phrase. With RTA strategy, grounding module learns region-level attribute association by aligning heatmaps generated from CLIP-based module. 🔥 means the parameters of the backbone are trainable, ❄ the parameters of the backbone are frozen.**

require collecting bounding box annotations for seen categories or enormous image-text pairs.

## 3 METHODOLOGY

### 3.1 Problem Formulation

Given an image $I$ and a noun phrase $T$, the task of PG requires the model to predict a bounding box $B$. For obtaining the box, a heatmap $H$ is also generated as an intermediate bridge. In zero-shot PG under weak supervision, the model grounds both seen and unseen classes during the inference, after only being trained on seen classes without bounding box annotations in images.

### 3.2 Overview of Proposed Framework

The overview of our proposed framework is shown in Figure 2. The framework comprises the CLIP-based module and the grounding module. The CLIP-based module (Figure 2 b) is built on CLIP, which is pre-trained on 400 million image-phrase pairs [37]. It first extracts region-level image semantics by transferring the textual class token's semantics to visual patch tokens. Then, this module fuses these tokens, generating a CLIP-based heatmap $A_C^*$.

The grounding module (Figure 2 c)) consists of an image encoder $\mathcal{E}_{img}(\cdot)$, a text encoder $\mathcal{E}_{txt}(\cdot)$, and a grounding decoder $\mathcal{D}_{gnd}(\cdot)$. Thus, the grounding module returns a heatmap $H$ as follows,

$$H = \mathcal{D}_{gnd}\left(\mathcal{E}_{img}(I), \mathcal{E}_{txt}(T)\right) \tag{1}$$

where the image encoder uses the last layer of the pre-trained CNN in ImageNet as the visual embedding. The text encoder uses the text

embedding branch of CLIP (VIT-B/32), which is frozen. Multimodal feature fusion calculates the similarity of text features and visual ones, $A_M = \mathcal{E}_{Img}(I) \otimes \mathcal{E}_{Txt}(T)$. The attention is then given as $R_M = \mathcal{E}_{img}(I) \circ A_M$, in which the symbol $\circ$ means Hadamard product. The grounding decoder converts the high-dimensional fusion features into the grounding heatmaps $H$. The decoder consists of two up-sampling layers.

Moreover, we propose triple alignment strategies, RTA, DomA, and CatA. RTA strategy helps align the grounding heatmaps $H$ with CLIP-based heatmaps $A_C^*$. DomA strategy aligns features of seen categories in the training and those in the pre-training. CatA strategy aligns phrases with the corresponding regions. Subsequently, we describe the three strategies in detail.

### 3.3 Region-text Alignment (RTA) Strategy

To learn region-level attribute associations, we exploit the working process of VLP. In our strategy, we employ the parameter-fixed CLIP to generate region-level attribute associations. Our network is trained on a limited number of seen classes. Therefore, directly re-training CLIP tends to overfit the seen classes as the model parameters are optimized only for seen classes. Consequently, knowledge learned for entity concepts unseen from the training set might be ignored during re-training. Parameter-fixed CLIP could potentially alleviate this issue.

Formally, for the CLIP image encoder, we denote the embeddings from the $l$-th transformer layer as $\{c^l, P^l\}$, where $c^l$ denotes the [CLS] token and $P^l = \{p_1^l, p_2^l, \cdots, p_Z^l\}$ denote the image patch

tokens. Inspired by VIT [38], the patch tokens of the last layer represent dense features. However, CLIP only focuses on the [CLS] token, and the patch tokens are unable to establish semantic associations with text embeddings [58]. A straightforward way is to transfer the [CLS] token's semantic information to the patch tokens in the last layer, discarding the preceding layers. Thus, each patch token only receives information from its corresponding semantic so that phrase-related visual entities are well grounded. In practice, we initialize the *mask token* from the pre-trained [CLS] token, i.e., $m^1 = c^1$ and append it to the above token sequence in the first layer of CLIP image encoder. Then the $l$-th transformer layer processes the mask tokens as follows,

$$m^{l+1} = \alpha^l \cdot \text{Attn}\left(m^l, P^l\right) + \beta^l \cdot m^l, \quad l = \{1, \cdots, L-1\} \quad (2)$$

where $A$ denotes the attention weight. Hyperparameters $\alpha$ and $\beta$ control the weights of the residual term. While the parameter $l < L - 1$, we set $\alpha$ and $\beta$ to avoid establishing strong semantic associations between the mask tokens and patch tokens in the shallow self-attention layers. These patch tokens tend to share holistic information, which is favorable in image-level tasks [47]. While the parameter $l = L - 1$, we enhance the semantic transfer from previous layer's mask tokens to the last layer's patch tokens. In other words, in shallow layers the two parameters are set relatively small, while in the last layer they are set relatively large.

**CLIP-based heatmap generation.** We obtain the final embedding vectors $\{m_p^L\}$, in which $p = 1, 2, \cdots, Z$ for each patch token. We then compute the inner product between the text embedding and the final embedding vectors to fuse texts and image regions. Formally, we obtain the CLIP-based heatmap as follows,

$$A_C = \exp\left(\frac{\mathcal{E}_{txt}(T) \otimes m_p^L}{\delta}\right) \quad (3)$$

where $\delta$ is the temperature scaling parameter in CLIP, $\mathcal{E}_{Txt}$ denotes text encoder, and $L$ represents the last layer of image encoder.

**CLIP-based heatmap refinement.** The gradient map is often used to ground specific objects [21, 45], where only the gradient of patch tokens of the last layer is computed. In contrast, we focus on patch-to-patch attention of each multi-head self-attention (MHSA). It involves collecting grounding-related features. Thus, we extract the patch-level attentions $A_{P2P} \in \mathcal{R}^{Z \times Z}$ based on patch tokens, without considering the [CLS] token. The gradient map $\nabla A$ is computed as $\nabla A = \frac{\partial Q}{\partial A_{P2P}}$, where $Q$ is the model's output logit, i.e., the similarity score for the image-text pair. The refinement is formulated as:

$$A_C^* = A_C \cdot \text{ReLU}\left(\sum_l \nabla A^l\right) \quad (4)$$

where $A_C^*$ is the CLIP-based heatmap refined by the gradient map.

**Region-level attribute association learning.** CLIP can associate attributes and entities. For example, CLIP can infer the image has "red elephant", though it has only seen the attribute "red" or the entity "elephant" during pretraining. Our CLIP-based heatmap $A_C^*$ inherits CLIP's property of attribute association. This is demonstrated in the image region (See Figure 3). To enable the grounding module to learn the similar property, smooth $\ell_1$ loss [39] forces the grounding module to simulate the region-text alignment process

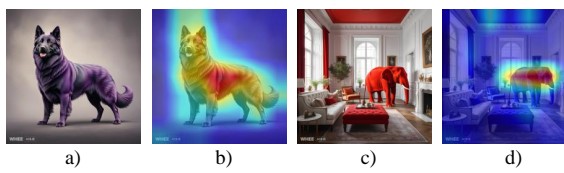

a)        b)        c)        d)

**Figure 3: Two visualization examples. a) and c) show two images generated by Generative AI [57]. a) is for "purple dog" and c) is for "red elephant". CLIP has never seen them before. b) and d) show our CLIP-based heatmaps.**

applied to the CLIP-based heatmap. The $\ell_1$ loss is given as

$$\ell_1(x) = \begin{cases} \frac{1}{2}x^2 & \text{if } |x| \leq 1 \\ |x| - \frac{1}{2} & \text{otherwise} \end{cases} \quad (5)$$

in which, $x$ is the difference between the grounding heapmatp $H$ and the CLIP-based heatmap $A_C^*$.

## 3.4 Domain Alignment (DomA) Strategy

To transfer knowledge from CLIP, a direct method is to replace the image encoder in our grounding module with that of CLIP. Then we train the modified model on the image-text datasets containing phrases from a limited number of classes. However, such a choice might cause severe overfitting.

This motivates us to seek a special kind of feature in the grounding module. Thus, we design DomA strategy: 1) Reconstruct grounding-related feature in CLIP and our network. 2) Minimize the difference in feature distributions between the training and pretraining phases. Formally, our grounding module computes the matching attention between a phrase and an image by

$$(A_M)_{i \times j \times 1} = \sum_{k=1}^{K} \mathcal{E}_{img}(I)_{i \times j \times k} \otimes \mathcal{E}_{txt}(T)_{1 \times k} \quad (6)$$

where $k$ and $i \times j$ donate the number of channels and spatial dimensions, respectively. The domain heatmap $A_M$, compared with the grounding heatmap $H$, can avoid overfitting. We consider the domain heatmap $A_M$ and CLIP-based heatmap $A_C^*$ as features, respectively. Note that the weakly-supervised training does not change the feature distribution of the frozen pre-trained model.

**Domain Alignment Loss.** To narrow the difference in feature distributions between the training and pre-training phases, We use contrastive learning [15]. We formulate an alignment process to learn to select the positive samples from a set of positive and negative matching attention maps. Specifically, we use the CLIP-based attention $A_C^*$ as the criterion for determining positive and negative samples. We treat the matching attention from the same input as positive samples, while the matching attention from different inputs as negative samples. Given a pair of matching attention $A_M$ and CLIP-based heatmap $A_C^*$, the alignment loss is given as

$$\begin{cases} \ell_\xi^{con}(A_M, A_C^{*\xi}) = \begin{cases} -\log \delta(1 - (A_M \cdot A_C^{*\xi})), \xi \in \mathcal{B}, \\ -\log \delta(A_M \cdot A_C^{*\xi}), \quad \xi \in \mathcal{W}, \end{cases} \\ \ell_{con}(A_M, A_C^*) = \frac{1}{|\mathcal{B} \cup \mathcal{W}|} \sum_{\xi \in \mathcal{B} \cup \mathcal{W}} \ell_\xi^{con} \end{cases} \quad (7)$$

where $A_C^*$ is a given in Eq. (4). $\mathcal{B}$ and $\mathcal{W}$ denote sets of positive and negative samples, respectively. $\mathcal{B} \cup \mathcal{W}$ is the union of two sets, and $\delta$ is the sigmoid function. Thus, our domain alignment is relevant to grounding-related domain generalization.

## 3.5 Category Alignment (CatA) Strategy

We treat the PG as a problem of phrase-region alignment. Inspired by the class activation method [27, 53], we use the phrase embedding $\mathcal{E}_{txt}(T)$ to discriminate the category of grounding-related feature, and normalize it as a category label $y$. To construct region-category relations, we calculate CLIP matching scores based on phrase embeddings and grounding-region embeddings. To ensure each training class can be seen by pre-trained CLIP, we crop out the grounding regions along the bounding box.

**Object and Phrase matching loss.** By exploiting the information from the phrase-related object by CLIP, the region of the object $B(H)$ is grounded by cropping the bounding box, reshaped as output $H$ size, and mapped to the image representation by CLIP image encoder, i.e., $v_O = \mathcal{E}_{img}(B(H))$. The cosine similarity between the object representation $v_O$ and the phrase representation $\mathcal{E}_{txt}(T)$ is used for a loss,

$$\ell_{OP} = -\sum_{n=1}^{N} y_n \log s_n, \tag{8}$$

in which $s_n$ is the cosine similarity. $N$ is the total number of generated heatmaps. Thus, the heatmap $H$ is gradually close to the phrase-related object under the supervision of $\ell_{OP}$.

**Random region and Phrase matching loss.** To enlarge the distance between phrase-irrelevant regions and phrase representations, the region of non-object is grounded by cropping a randomly generated $H$-size of the box $B_R$. The generated box should be less intersected with the bounding box $B(H)$. The box is represented by the CLIP image encoder, i.e., $v_R = \mathcal{E}_{img}(B_R)$. The cosine similarity between the box visual representation $v_R$ and the phrase representation $\mathcal{E}_{txt}(T)$ is used for a loss,

$$\ell_{RP} = -\sum_{n=1}^{N} y_n \log(1 - s_n^*), \tag{9}$$

in which $s_n^*$ denotes the cosine similarity. Thus, the heatmap $H$ retains less of the phrase-irrelevant region of the object.

**Regularization loss.** To further exclude irrelevant background in the heatmap, we constrain the size of the region covered by the heatmap, i.e.,

$$\ell_{RE} = \frac{1}{N} \sum_{n=1}^{N} H_n \tag{10}$$

Thus, the total loss of our model is given as follows,

$$\ell_T = \ell_1 + \lambda_1 \ell_{OP} + \lambda_2 \ell_{RP} + \lambda_3 \ell_{RE} + \lambda_4 \ell_{con} \tag{11}$$

Based on the grounding module, we generate the bounding box as follows. First, we set zeroes for the low-value pixels with a threshold of 0.5. Then, we search for contours and extract suitable bounding boxes following [46]. We then calculate the scores of bounding boxes based on the area percentage of the heatmap $H$. Finally, non-maximal suppression is applied with $IoU = 0.3$, and the boxes $B$ with 50% less than the maximum score are filtered

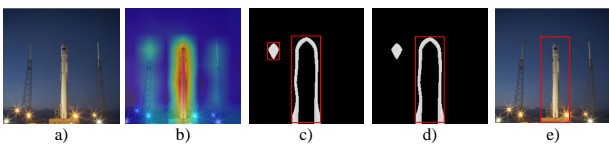

**Figure 4: Visualization of the bounding box generation. The phrase for the image is "white rocket". a) the input image, b) heatmap, c) contour map after thresholding with proposals, d) contour map after thresholding with the final bbox, and e) the input image with the final bbox.**

to complete the localization. The visualization of bounding box generation is shown in Figure 4.

## 4 EXPERIMENT

### 4.1 Datasets

We evaluate our framework on zero-shot PG settings using Flickr-Split-S0, Flickr-Split-S1, VG-Split-S2, and VG-Split-S3 [40]. In addition, to compare with previous weakly-supervised grounding methods, we use the setting in MG [1], which is adopted in various works using either MS-COCO [26] or VG [20] training splits, respectively. In both cases, the resulting models are evaluated on the testing splits of Flickr30K [35], VG, and ReferIt [6, 13].

### 4.2 Baselines and Metrics

We compare our framework with various state-of-the-art PG baselines, which can be divided into the following three categories. **1)** Supervised zero-shot baseline, i.e., ZSGNet [40]. **2)** Zero-shot baselines, including detector + CLIP [45], GAE [4], Grad-CAM [45], AdaptingCLIP [22], and MaskCLIP [58]. **3)** Weakly-supervised baselines, including MG [1], GbS [2], WWbl [41], SMST [42], BBR [12], and VPT [25].

Three metrics, including "pointing game" accuracy [55], bounding box accuracy [41], and recognition accuracy are used for our evaluation. The "pointing game" accuracy measures the percentage of predicted maximum points of the heatmap that lie within the bounding box ground truth. The bounding box accuracy measures the percentage of heatmap bounding boxes that have an IoU greater than 0.5 for the testing set of "image-query" pairs. In addition, the recognition accuracy counts the rate of correct results over all test sets. The accuracy relies on human evaluation. We asked volunteers to judge whether the grounded region is cognitively appropriate.

### 4.3 Implementation Details

For a fair comparison, we use VGG-16 as the visual encoder in our framework. The model accepts an image size of $224 \times 224$ which is the input size of CLIP's visual branch VIT-B/32. It generates a heatmap $H$ of the same size. We trained 150 epochs using SGD optimizer (a batch size of 64 and an initial learning rate of 0.0003), where the optimizer momentum is 0.9 and the weight decay is 0.0001. In addition, the layer $L$ is set as 11. The parameters $\alpha$ and $\beta$, they are set as 0.01 and 1 when $l < 11$. They are set as 1 and 10 when $l = L$. All methods were implemented on an NVIDIA RTX A6000. In all our experiments, the weights of our loss in Eq. (11) were set as follows, $\lambda_1 = 0.25$, $\lambda_2 = 0.125$, $\lambda_3 = 0.25$, and $\lambda_4 = 1$.

| Method | Visual Encoder | Flickr-Split-0 | Flickr-Split-1 | VG-2B | | VG-2UB | | VG-3B | | VG-3UB | |
|---|---|---|---|---|---|---|---|---|---|---|---|
| | | | | *0.3* | *0.5* | *0.3* | *0.5* | *0.3* | *0.5* | *0.3* | *0.5* |
| Supervised SoTA method [40] | VGG-16 | 39.32 | 29.35 | 17.09 | 11.02 | 16.48 | 10.55 | 17.63 | 11.42 | 17.35 | 10.97 |
| **VLP-based method** | | | | | | | | | | | |
| Detector+CLIP [45] | Faster RCNN | 14.12 | 12.60 | 11.78 | 4.82 | 10.69 | 4.77 | 12.57 | 5.74 | 11.68 | 4.67 |
| MaskCLIP [58] | ResNet-50 | 31.08 | 25.78 | 14.71 | 5.72 | 13.66 | 6.35 | **15.68** | 6.09 | 14.63 | 6.57 |
| Grad-CAM [45] | VIT-B/32 | 27.07 | 24.09 | 13.94 | 5.99 | 13.14 | 5.55 | 13.23 | 6.37 | 13.35 | 5.47 |
| GAE [4] | VIT-B/32 | 27.18 | 24.12 | **14.94** | 6.02 | 13.21 | 5.67 | 14.91 | 6.56 | 13.63 | 5.98 |
| AdaptingCLIP [22] | VIT-B/32 | 27.47 | 24.81 | 13.43 | 6.91 | 12.50 | 5.21 | 14.21 | 7.24 | 13.43 | 5.49 |
| **Weakly-supervised method** | | | | | | | | | | | |
| WWbl [41] | VGG-16 | 29.15 | 24.23 | 10.90 | 5.67 | 10.31 | 5.18 | 11.17 | 5.93 | 10.55 | 5.43 |
| **Ours** | VGG-16 | **32.50** | **28.02** | 14.12 | **6.92** | **13.74** | **6.45** | 14.57 | **7.48** | **14.83** | **6.61** |

Table 1: Bounding box accuracy across unseen splits. For Flickr-Split-0 & 1, we use the bounding box accuracy with an IoU threshold of 0.5. For VG-Split-2 & 3, we report the bounding box accuracy with IoU thresholds of 0.3 and 0.5, respectively. "B" and "UB" denote the balanced and unbalanced sets in VG-Split, respectively.

| | Method | Point Accuracy | | | Bbox Accuracy | | |
|---|---|---|---|---|---|---|---|
| | | *VG* | *Flickr* | *Referlt* | *VG* | *Flickr* | *Referlt* |
| | GAE [4] | 54.72 | **72.47** | 56.76 | 16.70 | 25.56 | 19.10 |
| | **CH** | **55.31** | 71.29 | **57.47** | 23.43 | 43.75 | 24.63 |
| MS-COCO | MG [1] | 47.94 | 61.66 | 47.52 | 15.77 | 27.06 | 15.51 |
| | GbS [2] | 52.00 | 72.60 | 56.10 | - | - | - |
| | WWbl [41] | 59.09 | 75.43 | 61.03 | 27.22 | 35.75 | 30.08 |
| | SMST [42] | 62.96 | **78.10** | 61.53 | 29.14 | 46.62 | 32.43 |
| | BBR [12] | 60.05 | 77.19 | **63.48** | 28.77 | **47.26** | 30.63 |
| | **Ours** | 60.31 | 77.85 | 62.63 | **29.58** | 45.46 | **33.41** |
| VG | MG [1] | 48.76 | 60.08 | 60.01 | 14.45 | 27.78 | 18.85 |
| | GbS [2] | 53.40 | 70.48 | 59.44 | - | - | - |
| | WWbl [41] | 62.31 | 75.63 | 65.95 | 27.26 | 36.35 | 32.25 |
| | SMST [42] | **66.63** | **79.95** | 70.25 | 30.95 | 45.56 | **38.74** |
| | BBR [12] | 63.51 | 78.32 | 67.33 | **31.02** | 42.40 | 35.56 |
| | **Ours** | 58.07 | 76.69 | **70.86** | 27.31 | **45.63** | 35.70 |

Table 2: Comparison with SoTA weakly-supervised PG methods evaluated using the "pointing game" accuracy and bounding box accuracy on VG, Flickr30K, and ReferIt. The best performances are shown in bold.

## 4.4 Main Results

**Zero-shot Evaluation on Unseen Phrase Classes.** We report the zero-shot evaluation results on the test split of Flickr-Split and VG-Split in Table 1. Unlike Flickr-Split, VG-Split's phrases contain a large amount of textual noise that does not describe the corresponding objects. Therefore, we set two IoU thresholds (0.3 and 0.5) in the evaluation. Specifically, our approach achieves superior results on IoU thresholds of 0.5 shown in column #6, #8, #10, and #12). This indicates that the model's grounding results cover unseen categories better than other methods. Although there still exists a gap compared to the supervised method, our method improves the performance significantly compared to the weakly-supervised methods. WWbl model masks the image with the heatmaps, followed by using external knowledge to measure the similarity between masked images and phrases as the loss function. However, the external discriminator rarely encounters mask-covered images, leading to an incorrect accumulation of category judgment. Compared to other datasets, Flickr-Split-1 is more stringent in defining the difference between training phrase categories and testing ones.

However, our results are close to the supervised grounding SoTA on Flickr-Split-1. This result shows a strong generalization of our model on unseen phrase classes.

**Weakly Supervised Evaluation on Seen Classes.** We report the performances of our PG framework compared with other weakly-supervised PG methods on Flickr30K, VG, and ReferIt in Table 2. For a fair comparison, all trainable methods use VGG-16 as the image encoder to produce the final grounding output. The experimental results show that our method generates competitive results compared with other weakly-supervised PG methods. Our CLIP-based heatmap (CH) also surpasses the pseudo label (GAE) used by WWbl in terms of bbox accuracy and such information is not available in the training dataset. This explains the 9% increase in bbox accuracy over WWbl. Our method does not perform well on some metrics compared to BBR, and SMST. The main reason is that these methods provide more accurate supervision labels. SMST designs visual similarity maps, while BBR uses bounding boxes provided by the object detector as additional annotations. We did not consider using their labels because the zero-shot generalization of weakly-supervised PG is more important than the slight improvement in grounding accuracy. Although visual similarity maps and box annotations can improve the appearance of heatmaps, they are both text-agnostic. Simply using such annotations can disrupt the text-image alignment, which is crucial for zero-shot manner of weakly-supervised PG.

**Zero-shot Evaluation on Unseen Object Classes.** We compare our framework with the supervised methods on unseen image-object classes. The testing set was collected which contained 30 instances for each category from Google. These categories belong to novel concepts or fine-grained categories, which do not appear in training datasets. This dataset includes 10 categories: CElebrity names (CE), ANime names (AN), GAme character names (GA), ARtwork names (AR), RAre plant and animal phrases (RA), SMall object phrases (SM), EXclusive category phrases (EX), SEntence-level phrases (SE), REmote-sensing-related phrases (RE) and MEdical-related phrases (ME). Note that the supervised method uses box annotations during training, whereas our method does not require any form of additional annotations. Figure 5 shows the proportion of grounding results evaluated by human evaluation. The subjective results clearly show that our method outperforms the supervised

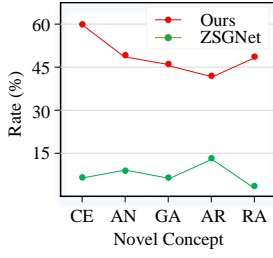
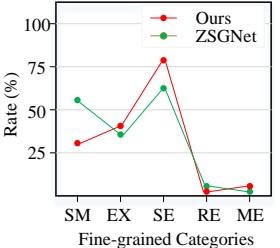

**Figure 5: Comparison of the recognition accuracy of different models with unseen object categories.**

| Variant | Strategy | | | |
|---|---|---|---|---|
| | RTA | DomA | CatA | R+D+C |
| CLIP-based Heatmap | 13.48 | 11.20 | 10.55 | 41.58 |
| Gradient map | 11.28 | 10.31 | 9.79 | 39.48 |
| Combination | 16.85 | 12.23 | 11.10 | 45.46 |

**Table 3: Grounding results with different alignment strategies on the val split of Flickr30K Entities.**

| | Method | Point Accuracy | | | Bbox Accuracy | | |
|---|---|---|---|---|---|---|---|
| | | VG | Flickr | Referlt | VG | Flickr | Referlt |
| MS-COCO | $\ell_1$ | 49.17 | 58.11 | 46.72 | 13.01 | 16.85 | 16.56 |
| | $\ell_1+\ell_{con}$ | 51.68 | 58.59 | 46.97 | 14.32 | 29.41 | 18.83 |
| | $\ell_1+\ell_{con}+\ell_{OP}$ | 53.31 | 70.54 | 48.33 | 20.93 | 38.25 | 25.04 |
| | $\ell_1+\ell_{con}+\ell_{OP}+\ell_{RP}$ | 55.20 | 73.34 | 60.92 | 23.82 | 42.51 | 29.09 |
| | $\ell_1+\ell_{con}+\ell_{OP}+\ell_{RP}+\ell_{RE}$ | 60.31 | 77.85 | 62.63 | 29.58 | 45.46 | 33.41 |
| VG | $\ell_1$ | 48.42 | 57.85 | 50.11 | 13.47 | 16.10 | 17.92 |
| | $\ell_1+\ell_{con}$ | 37.26 | 58.67 | 50.28 | 14.58 | 28.20 | 19.90 |
| | $\ell_1+\ell_{con}+\ell_{OP}$ | 51.97 | 70.12 | 51.47 | 20.11 | 36.59 | 27.88 |
| | $\ell_1+\ell_{con}+\ell_{OP}+\ell_{RP}$ | 53.98 | 73.28 | 64.62 | 23.44 | 40.79 | 31.98 |
| | $\ell_1+\ell_{con}+\ell_{OP}+\ell_{RP}+\ell_{RE}$ | 58.07 | 76.69 | 70.86 | 27.31 | 45.63 | 35.70 |

**Table 4: Comparison of the point accuracy and bbox accuracy of using different combinations of loss functions on MSCOCO and VG datasets.**

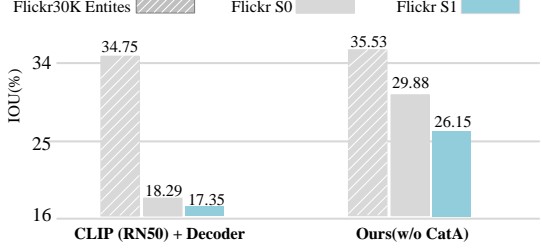

**Figure 6: Comparison with the baselines.**

method with large margins in novel concepts. Some fine-grained categories, including SM, EX, and SE with close semantics have been encountered during training. It is extremely advantageous to the supervised method because the box annotations build stronger associations among similar semantic entities.

## 4.5 Ablation Study

**Effects of alignment Strategies.** Firstly, Table 3 shows the effects of alignment strategies in weakly supervised settings. We use Flickr30K Entities as it contains clearer expressions without noise. Among all combinations of VLP and training stages, using all alignment strategies leads to the best performance. Secondly, we measure the impact of each loss on grounding performance. Table 4 shows quantitative comparisons among different combinations of loss functions. Our framework only obtains 16.85% when only $\ell_1$ is used. An addition of $\ell_{con}$ improves IoU from 16.85% to 29.41% and the inclusion of $\ell_{OP}$ and $\ell_{RP}$ achieves absolute improvements 8.84% and 4.26%, respectively. $\ell_{RE}$ can ensure the compactness of CLIP-based heatmaps and improve the IoU by 2.95% on the dataset. Thirdly, we compare the baseline (we replaced the image encoder of grounding module with CLIP (RN50)) with our domain alignment strategy. In our experiments, we evaluate seen classes on Flickr30K Entities, and unseen classes on Flickr S0 and Filckr S1. As shown in Figure 6, our baseline has a large gap in bounding box accuracy on seen and unseen classes. Compared to the former, our domain alignment strategy achieves better domain adaptation between seen and unseen classes. Finally, we ablate each alignment strategy in zero-shot settings. In addition, Table 5 shows the quantitative comparisons among different combinations of alignment strategies. The results show that each strategy works in our PG framework.

**Effects of CLIP-based Heatmap.** To show the impact of the pseudo-label effect on our grounding module, we show the performance of our CLIP-based heatmap in Table 2 and Table 6, and evaluate the performance of our framework with different training labels (rows #5 and #6 in Table 6). Our CLIP-based heatmap has demonstrated competitive performance in seen classes. In addition, when our CLIP-based heatmap (CH) acts as a label, our network improves the grounding quality in most seen classes.

## 4.6 Qualitative Analysis.

We show several results of our alignment strategies in Figure 7. We use the instances' ground-truth bounding boxes as proposals in Flickr30K. When using only RTA, the predicted CLIP-based heatmap tends to focus on the most discriminative region of the referred object. However, when equipped with DomA strategy, the predicted CLIP-based heatmap tends to capture the context of the phrase but may focus on different object categories. With triple alignments, our method successfully grounds the referred object.

Furthermore, we compare our method with VLP-based and weakly-supervised PG methods, as shown in Figure 8. We observe that our method typically grounds more complete object contents and less phrase-related background regions. Specifically, VLP-based methods and weakly-supervised PG methods may underestimate the region of blue jeans and the woman, or falsely ground the region of mountain bike and the subway station. In contrast, the regions grounded by our framework are more complete and compact.

Finally, we show failure cases of our framework in Figure 9. We categorize failure cases into two groups: similar dense objects and in-context entities-related objects. Our framework highlights connected regions rather than separate regions while it locates dense objects. The number of bounding boxes cannot be precisely determined. Furthermore, our framework extracts only noun phrases

| Method | Strategy | Flickr-Split-0 | Flickr-Split-1 | VG-2B | | VG-2UB | | VG-3B | | VG-3UB | |
|---|---|---|---|---|---|---|---|---|---|---|---|
| | | | | S0 | S1 | S0 | S1 | S0 | S1 | S0 | S1 |
| $\ell_1$ | RTA | 28.32 | 25.01 | 10.67 | 5.51 | 10.23 | 5.16 | 11.24 | 5.87 | 10.64 | 5.41 |
| $\ell_1 + \ell_{con}$ | RTA+DomA | 29.88 | 26.15 | 11.33 | 5.69 | 11.25 | 5.39 | 12.08 | 6.12 | 11.30 | 5.65 |
| $\ell_1 + \ell_{con} + \ell_{OP}$ | RTA+DomA+CatA | 30.58 | 26.99 | 12.07 | 5.94 | 12.12 | 5.81 | 12.73 | 6.54 | 12.66 | 5.93 |
| $\ell_1 + \ell_{con} + \ell_{OP} + \ell_{RP}$ | RTA+DomA+CatA | 31.33 | 27.74 | 13.85 | 6.39 | 13.05 | 6.06 | 13.15 | 7.03 | 13.55 | 6.42 |
| $\ell_1 + \ell_{con} + \ell_{OP} + \ell_{RP} + \ell_{RE}$ | RTA+DomA+CatA | **32.50** | **28.02** | **14.12** | **6.92** | **13.74** | **6.45** | **14.57** | **7.48** | **14.83** | **6.61** |

**Table 5: Bounding box accuracy across unseen splits. For Flickr-Split-0&1 we use accuracy with IoU threshold of 0.5. For VG-Split-2&3, we report accuracy with IoU thresholds of 0.3 and 0.5. "B" and "UB" are balanced and unbalanced sets in VG-Split.**

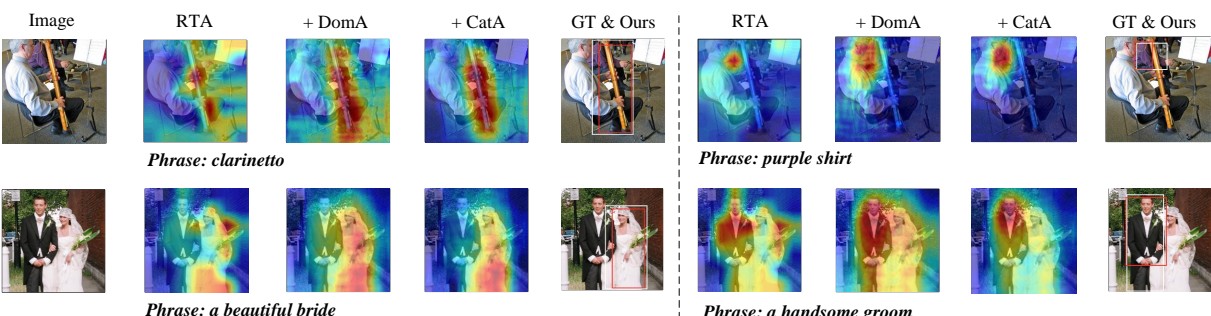

**Figure 7: Qualitative results are reported. Input Images are given in the left most column. The corresponding phrases are different. Columns #2-4 present the generated heatmaps using RTA, RTA+DomA, and triple alignments, respectively. Columns #6-8 show the results of another phrase. The white boxes represent ground truth and the red represents the results of ours.**

| Method | Overall | People | Animals | Vehicles | Scene | Other |
|---|---|---|---|---|---|---|
| MaskCLIP | 34.26 | 37.46 | 40.93 | 52.25 | 48.40 | 25.87 |
| AdaptingCLIP | 29.47 | 29.23 | 40.15 | 45.00 | 41.86 | 24.92 |
| GAE | 25.56 | 26.76 | 39.72 | 38.12 | 33.72 | 22.22 |
| CH | 43.75 | 56.33 | 62.31 | 58.60 | 52.78 | 32.26 |
| Ours w/ GAE | 36.35 | 43.58 | 48.22 | 52.72 | 55.94 | 26.44 |
| Ours w/ CH | **45.46** | **56.44** | 59.95 | 57.68 | **70.04** | **32.53** |

**Table 6: Category-wise bounding box accuracy on Flickr30K Entities. Boldface: best results. Underline: suboptimal results.**

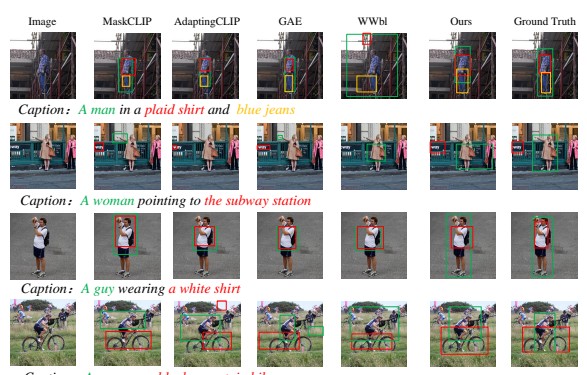

**Figure 8: Qualitative results of our framework with other methods. Each bounding box represents the object region referred by specific noun phrase.**

without considering phrases in-context during the inference. This leads to an inaccurate evaluation of the referred object's location.

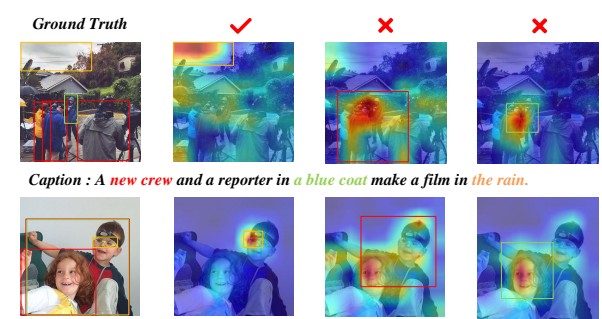

**Figure 9: Failure cases of our method. Column #1 presents the ground truth. Column #2 presents the successful case. Column #3 presents the failure cases for grounding similar dense objects. Column #4 presents the failure cases for grounding entities in context.**

## 5 CONCLUSION AND FUTURE WORK

In this paper, we propose a PG framework, which designs alignment strategies to address three key problems of zero-shot grounding with weak supervision. Our approach outperforms previous zero-shot methods and achieves competitive results on weakly supervised benchmarks.

In the future, there are several interesting directions. For instance, we will consider interpretable solutions for several grounding-related tasks, such as Grounded VQA and image captioning. In addition, we will consider how to use multi-modal large language models [36] to improve zero-shot PG under weak supervision.

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
