# OpenReview forum: "Triple Alignment Strategies for Zero-shot Phrase Grounding under Weak Supervision"
_acmmm.org/ACMMM/2024/Conference — MM2024 Poster_

### Official Review · Reviewer_HDdX · 2024-05-22

**Rating:** 3
**Confidence:** 4

**Summary:**

This paper studies Phrase Grounding (PG) under both settings of weak supervision and zero-shot. It alleviates three issues: attribute association, knowledge transfer, and category sim & diff measurement, by triple alignment strategies, that is, region-text alignment (RTA), domain alignment (DomA), and category alignment (CatA). Experiments are carried out on FlickrSplit-S0, Flickr-Split-S1, VG-Split-S2, and VG-Split-S3 to show effectiveness.

**Strengths:**

[+] Zero-shot phrase grounding under weak supervision is more practical in reality, thus embarking on the right path.

[+] The paper is easy to follow and understand.

[+] Many experiments are conducted to prove the effectiveness of core components, including loss functions, heatmap fusion, several strategy ablations.

**Limitations:**

[-] Sub-optimal designs. This paper uses heatmaps from frozen CLIP as guidelines without training, since limited training data makes it easy to overfit. However, fine-tuning CLIP using small amount of data has been widely-used recently, such as adapter, LoRA, and prompt tuning. Comparing to these fine-tuning operations, is triple alignment strategies (this paper) better? Please make comprehensive discussions and experiments.

[-] Sub-optimal performance. In Tab.1-2, this paper obtains moderate results, sometimes it is actually worse than VLP-based methods that does not require fine-tune CLIP. Also, in pursuit of fair comparison, please quantify training parameters, training/testing time cots in these tables.

[-] Special cases. In introduction section (Fig.1), this paper emphasizes attribute association and knowledge transfer, the corresponding visualization or statistical analysis should be reported in experimental results.

**Suitability:**

3

---

### Official Review · Reviewer_6cu3 · 2024-05-25

**Rating:** 3
**Confidence:** 3

**Summary:**

This paper presents a novel framework for Phrase Grounding (PG) that operates under weak supervision and zero-shot settings. The framework leverages three alignment strategies: region-text alignment (RTA), domain alignment (DomA), and category alignment (CatA). The proposed approach aims to bridge the gap between seen and unseen categories during training by aligning visual and textual information at different levels, ensuring robust performance in real-world applications without requiring extensive bounding box annotations.

**Strengths:**

1. The exploration of zero-shot phrase grounding with weak supervision is meaningful.
2. The proposed DomA strategy can effectively avoid overfitting.

**Limitations:**

1. Lack of comparison with other weakly-supervised models in the main table. As mentioned in 4.2, the author states that they compare their methods with several weakly-supervised methods. However, in Table 1, only WWbl is involved in the comparison. In Table 2, the performance of the proposed method is not satisfactory. Therefore, more demonstration is needed to justify the performance.
2. Comparisons with ZSGNet in Figure 5 are not persuasive enough. In this work, the author adopts CLIP to guide the model. CLIP is much more powerful in zero-shot prediction. Therefore, there is a need to have experiments compared to the VLP-based model to demonstrate the ability of the model on unseen objects.
3. The motivation of DomA is not clear enough and its effectiveness is not well demonstrated. In the introduction, the author states that DomA aims to transfer knowledge from seen classes. Meanwhile, in the method, they state that DomA is used to narrow the feature distribution of training and pre-training. It is a little bit confusing. On the other hand, the effectiveness of the DomA is not well demonstrated. It would be interesting to see the model accuracy on seen and unseen classes with and without DomA.
4. In Table 3, it is strange that the three independent strategies have poor results, but together they do perform well. The experiment results need more explanation.

**Suitability:**

3

---

### Official Review · Reviewer_JCyi · 2024-06-07

**Rating:** 4
**Confidence:** 3

**Summary:**

This paper proposes propose a framework of zero-shot phrase grounding under weak supervision, which is built on triple alignment strategies, including region-text alignment strategy, domain alignment strategy and category alignment strategy. Experimental results on zero-shot and weakly-supervised phrase grounding show the effectiveness of proposed method.

**Strengths:**

1-It makes a research on phrase grounding task in weak supervision and zero-shot settings at same time, and proposes a zero-shot phrase grounding framework.

2-The proposed method designs region-text alignment strategy based on CLIP, and tries to minimize the difference between distributions of seen classes by narrow the difference in feature distributions between the training and pre-training phases.

3-It considers both category semantics and region-category relations at same time, which regards PG as a problem of phrase-region alignment.

**Limitations:**

1-In line 101, it mentions that VLP-based methods ground objects on new data without being fine-tuned. But in lin 105, the paper says that these VLP-based methods suffer from the collecting of enormous training data. When using pretrained VLP model, why it suffers from the enormous training data.

2-It seems like Zero-shot Referring Expression Comprehension, whether the proposed method is also effective?

3-In Table 2, compared with SMST and BBR, the proposed method seems limited. Please analyse the main reason. Whether can provide only in weak supervision setting or zero-shot setting.

4-In 3.3, the CLIP image encoder has been used (Fig-2-b), why in line 569 VGG-16 is used as the visual encoder (Fig-2-c). Whether exists multiple visual encoder? If the CLIP has been used in 3.3, why do not also use CLIP to replace VGG-16.

5-In fact, as shown in Fig-2-c, maybe the multimodal feature fusion is more effective if using CLIP visual encoder and CLIP text encoder. Please provide the related results in above settings, and analyse the reason.

6-For the BBox Generation & Crop (Fig-2-d), the feature map focuses on the motorcycle, thus the image is also corresponding it, but not both motorcycle and man, please refine this part for clearly understanding.

**Suitability:**

3

---

### Meta-Review · Area_Chair_9H1J · 2024-07-01

**Recommendation:** Accept (Poster)
**Confidence:** 3

**Metareview:**

The AC goes through the paper, rebuttal and review comments. This paper got 3 borderline accept reviews. All reviewers acknowledge the motivation and novelty. Therefore, the AC thinks this paper can be accepted, but the authors should carefully improve the paper by leveraging the feedback.